# Vitamin D Deficiency as a Risk Factor of Preeclampsia during Pregnancy

**DOI:** 10.3390/diseases11040158

**Published:** 2023-11-02

**Authors:** Chrysoula Giourga, Sousana K. Papadopoulou, Gavriela Voulgaridou, Calliope Karastogiannidou, Constantinos Giaginis, Agathi Pritsa

**Affiliations:** 1Department of Nutritional Sciences and Dietetics, School of Health Sciences, International Hellenic University, 57400 Thessaloniki, Greece; chrysagiourga@gmail.com (C.G.); sousana@the.ihu.gr (S.K.P.); gabivoulg@gmail.com (G.V.); karasto@ihu.gr (C.K.); 2Department of Food Science and Nutrition, School of Environment, University of the Aegean, 81400 Myrina, Greece; cgiaginis@aegean.gr

**Keywords:** vitamin D, preeclampsia, pregnancy, nutrition, review

## Abstract

A balanced diet is achieved not only via the consumption of a variety of food products but also by ensuring that we take in sufficient quantities the micronutrients necessary for the adequate functioning of the human body, such as vitamins, an important one of which is vitamin D. Vitamin D has been closely linked to bone health. Vitamin D deficiency has often been associated with negative effects concerning several pregnancy adverse outcomes, the most important of which are the birth of SGA newborns, premature birth, and, finally, preeclampsia, which are discussed in this work. The aim of this review is to critically summarize and scrutinize whether the concentration of vitamin D in the blood serum of pregnant women in all its forms may be correlated with the risk of preeclampsia during pregnancy and whether vitamin D levels could act both as a protective agent or as a risk factor or even a prognostic measure of the disease. The association of vitamin D levels with the onset of preeclampsia was examined by searching the PubMed and Google Scholar databases. A total of 31 clinical trials were identified and included in this review, with the aim of summarizing the recent data concerning vitamin D levels and the risk of preeclampsia. Among them, 16 were published in the last five years, and 13 were published within the last a decade. Most studies showed a significant association between vitamin D deficiency and preeclampsia risk. It was also found that the higher the dose, the lower the risk of disease. Of the 31 articles, only 7 of them did not show a significant difference between vitamin D levels and preeclampsia regardless of comorbidity. The results of this review suggest that there is indeed an association between the concentration of vitamin D during pregnancy and the risk of preeclampsia; however, further studies are strongly recommended to derive conclusive evidence.

## 1. Introduction

The term “vitamin D” was officially used in 1922 [1] to denote a vitamin essential for promoting calcium deposition in the human body. This discovery stemmed from the study of a disease that afflicted individuals as early as 1822 [2,3], rickets, a condition that leads to weakened bones [4]. The term “rickets” itself was first documented in 1634 [5]. It was later determined in 1651 that rickets could also result from insufficient sun exposure [6,7]. Both sun exposure and cod liver oil could serve as potential treatments for rickets, effectively preventing its occurrence [8,9,10,11,12].

Vitamin D is a fat-soluble vitamin and steroid hormone [4,12,13] primarily synthesized in the skin through exposure to ultraviolet radiation from the sun (UVB radiation) [14]. This essential nutrient exists in two forms: ergocalciferol (vitamin D2), primarily found in plant-based foods [15], and cholecalciferol (vitamin D3) [16,17], mainly present in animal-derived foods [16] and synthesized in the skin through sun exposure [18], originating from a precursor similar to cholesterol [4]. Vitamin D deficiency remains widespread [19] primarily due to lifestyle factors [20]. During pregnancy, vitamin D deficiency can lead to adverse outcomes such as preeclampsia, gestational diabetes mellitus (GDM), and detrimental effects on the health of the infant [21,22].

### 1.1. Preeclampsia

According to the American College of Obstetrics and Gynecology, preeclampsia is defined as the presence of hypertension and proteinuria occurring after the 20th week of pregnancy [23,24,25,26]; it was identified prior to the 34th week of pregnancy in a pregnant women who previously had normal blood pressure levels [25]. The diagnosis of preeclampsia can be challenging, as proteinuria is often one of the last detected features, leading to delays in diagnosis and subsequent management. It is essential to consider several criteria before diagnosing this disease.

Preeclampsia can also manifest as a disorder of the placenta [27]. Placental preeclampsia, a severe form of this condition [28,29], is categorized into two stages [23]: the first stage involves abnormal placental development during the initial trimester of pregnancy, while the second stage occurs during the last two trimesters, characterized by an excess of several anti- angiogenic factors [27].

The management of preeclampsia necessitates pre-pregnancy counseling, continuous screening throughout pregnancy, and regular blood pressure monitoring even after delivery. Antenatal aspirin and magnesium therapy may be recommended for a women meeting the ‘high risk’ classification criteria. It is crucial to monitor for preeclampsia even in pregnant women displaying no apparent symptoms. Strict blood pressure control can significantly reduce the prevalence of this condition [27].

Hypertensive disorders during pregnancy vary in severity and may include chronic hypertension with systolic blood pressure ≥140 mmHg or diastolic blood pressure ≥90 mmHg [30]. Preeclampsia poses a substantial risk to pregnant women, contributing to a significant number of maternal deaths, making it one of the most important complications during pregnancy [31,32,33]. The prevalence of this disease varies worldwide, ranging from 2 to 8% in developed countries [18,30,34,35,36,37,38] or even more [39]. Notably, the incidence of preeclampsia during pregnancy adversely affects newborns, potentially resulting in small-for-gestational-age neonates or premature delivery [40]. This emphasizes the urgency of understanding, diagnosing, and effectively managing preeclampsia to ensure the well-being of both mothers and their infants.

### 1.2. Pathophysiology of Preeclampsia

The pathophysiology of this pregnancy-related disorder remains largely unexplained, as fully elucidating its mechanisms and etiology has proven challenging. However, it involves several key factors, including abnormal placental development, spiral artery abnormality [27,41,42,43,44], placental insufficiency, and endothelial dysfunction [27]. The analysis of placentas from women with preeclampsia often reveals arterial sclerosis, leading to inadequate perfusion and subsequent ischemia [27,45,46].

During preeclampsia, abnormal vascular response to placentation triggers placental hypoxia, contributing to placental insufficiency [27,47]. In contrast, a normal first-trimester pregnancy experiences low oxygen levels, leading to the production of the angiogenic factor HIF-1α [27]. Hypoxia conditions in these circumstances are regulated by 2-methoxy estradiol, which blocks the expression of HIF-1a [48]. In cases of hypoxia, sFlt-1 conversion occurs due to fluctuations in HIF-1α production, resulting in endothelial dysfunction in preeclampsia [27].

Placental hypoxia plays a pivotal role in the pathophysiology of preeclampsia. In hypoxic conditions, nitric oxide synthesis is activated [37], leading to an increase in its levels [27,49]. Nitric oxide serves as a potent vasodilator, relaxing spital arteries and reducing vascular resistance. However, during preeclampsia, nitric oxide levels decrease, causing vasoconstriction. This reduction in blood flow to the placenta becomes critically significant in the progression of the disease. The intricate interplay of these factors underscores the complexity of preeclampsia’s pathophysiological mechanisms, necessitating further comprehensive research to better understand the disorder.

During pregnancy, oxidative stress in the placenta increases due to the mitochondria’s production of excess reactive oxygen species (ROS), a phenomenon exacerbated in preeclampsia due to abnormal placentation [50]. Notably, levels of essential antioxidants such as vitamin C, A, and E, along with iron binding, are diminished in women with preeclampsia [27].

The role of angiogenic factors in preeclampsia development is crucial, as adequate angiogenesis is essential for normal fetal growth. Any disruption in vascular endothelial growth factor (VEGF) expression, function, or receptor activity can adversely impact the endothelium. VEGF levels significantly decrease in preeclampsia, while sFlt-1 levels rise, interfering with the normal flow of VEGF [27,49]. Flt-1 acts as a receptor for VEGF and can disrupt its normal flow. The expression patterns of VEGF, placental growth factor, and VEGF-1 are altered in preeclampsia, with placental growth factor, a vital angiogenic protein for fetal angiogenesis, being notably reduced. This reduction is associated with the occurrence of preeclampsia. A pivotal mediator of preeclampsia is circulating soluble fms-type tyrosine kinase-1, which causes vasoconstriction and endothelial dysfunction in vivo [27,48,49]. Additionally, s-Endoglin, possessing antiangiogenic properties, binds to its receptors in the bloodstream, blocking the action of aforementioned factors [27,49]. Hence, the expression of placental angiogenic factors is vital for fetal development [27].

Some studies have implicated the production of autoantibodies in preeclampsia development. During pregnancy, angiotensin-II levels in the body are elevated, and this is necessary for the woman’s vasomotor response. In preeclampsia, angiotensin type-1 receptor increases placental vascular sensitivity, sequestering angiotensin-II and reducing cytotrophoblast recruitment. An increased expression of angiotensin receptor type-1, activated by plasminogen, impacts preeclampsia, leading to intermediate actions and reduced plasmin synthesis. Consequently, placental trophoblast invasion is hindered, and an observed immune imbalance significantly contributes to preeclampsia development [49].

### 1.3. Diagnosis of Preeclampsia

The diagnostic criteria for preeclampsia have been defined by the American College of Obstetricians and Gynecologists [25]. According to these guidelines, one of the criterion related to blood pressure involves a systolic blood pressure level of ≥140 mmHg and a diastolic blood pressure of ≥90 mmHg [40,51,52], measured on two separate occasions at least 4 h apart after the 20th week of pregnancy in a woman who previously had normal blood pressure. Another specific criterion is the presence of systolic blood pressure ≥ 160 mmHg and diastolic blood pressure ≥ 110 mmHg [25].

As previously mentioned, proteinuria is a significant symptom of the disease. The criteria for diagnosing proteinuria in the context of preeclampsia include the following: protein excretion of ≥300 mg/24 h in urine collection [25,34,53], a protein/creatinine ratio of ≥0.3 mg/dL, and a level indication of 1+ [54,55]. In cases where proteinuria is absent, new-onset hypertension can be diagnosed based on different criteria. These criteria encompass thrombocytopenia, renal failure characterized by serum creatinine concentrations >1.1 mg/dL, liver dysfunction, elevated liver transaminase levels, pulmonary edema [40], new-onset headache unresponsive to medication and not attributable to alternative diagnoses, or visual symptoms. These guidelines form the basis for the accurate and timely diagnosis of preeclampsia, ensuring appropriate medical intervention.

### 1.4. Vitamin D Sources

It is commonly known that only a limited range of foods qualify as rich sources of vitamin D [4]. Fatty fish, including sardines, tuna, mackerel, and salmon, as well as cod liver oil, are sources with high amounts of vitamin D [56]. Cod liver oil not only provides an ample amount of vitamin D but also serves as a valuable source of vitamin A and n3- fatty acids. Furthermore, eggs and their yolks, mushrooms, and liver are also recognized as sources of this essential nutrient.

However, it is crucial to emphasize that sunlight remains the primary source of vitamin D intake, accounting for 90% of the body’s daily vitamin D requirements [4]. It is essential for individuals to make proper food choices to ensure an adequate intake of vitamin D for optimal health.

### 1.5. Biological Role and Forms

The two primary forms of vitamin D, namely ergocalciferol (Vitamin D2) and cholecalciferol (Vitamin D3) [4], serve as precursors for the hormone responsible for regulating calcium metabolism and homeostasis. Cholecalciferol is synthesized from 7-dihydroxycholesterol through exposure to sunlight. These initial forms of vitamin D are biologically inactive and require further enzymatic processing to become active. Initially, hydroxylation occurs in the liver, producing calcidiol, also known as 25(OH)D [4]. Subsequently, in the kidneys, 1-alpha hydroxylation takes place, leading to the formation of the most potent form, calcitriol 1,25(OH)2D [4]. Notably, during early pregnancy, there is an increase in active 1,25-dihydroxyvitamin D3 [57].

This enzymatic process is significantly influenced by the parathyroid hormone (PTH) and other hormones, underscoring the potential autocrine–paracrine role of calcitriol [58,59]. The paramount function of vitamin D lies in promoting the cellular differentiation of enterocytes and facilitating the intestinal absorption of calcium, thereby contributing to calcium homeostasis. The parathyroid gland secretes PT, which stimulates 1-α-hydroxylation in the kidney, leading to the generation of calcitriol [4]. As the concentration of this specific form of vitamin D increases, so does calcium transport. Additionally, the activity of osteoblasts and osteoclasts is finally regulated. Once plasma calcium levels return to normal, PTH secretion decreases. This intricate interplay demonstrates the essential role of adequate vitamin D in maintaining normal blood calcium levels.

In vitamin D deficiency status, calcium absorption diminishes, leading to an increase in circulating parathyroid hormone. Therefore, it is recommended to evaluate both calcidiol and parathyroid hormone levels to accurately assess vitamin D status. This evaluation is crucial for understanding the intricate balance between vitamin D, calcium, and hormonal regulation in the body.

### 1.6. The Role of Vitamin D in Bone Health

Vitamin D plays a crucial role in maintaining bone health [60,61]. Vitamin D deficiency can lead to rickets in infants and children, as well as osteomalacia in adults in the later stages of life, although this condition is rare in developed nations [4]. Notably, vitamin D deficiency has also been associated with osteoporosis and an increased incidence of falls and fractures, mainly in older adults. The process of mineralization in bones commences early in pregnancy, with significant deposition occurring during the third trimester especially. Consequently, bone mass escalates dramatically, reaching up to forty-fold by adulthood, with 90% of peak bone mass attained by the end of an individual’s second decade of life [62,63]. It is imperative to recognize that both childhood and adulthood are critical periods for bone mineral deposition.

When dietary sources are limited and sun exposure is insufficient [64], oral supplementation is recommended. The effectiveness of oral vitamin D supplements in preventing falls and fractures remains incompletely understood. It has been shown that simultaneous calcium and vitamin D intake can reduce hip fractures [65]. However, it is worth noting that excessively high doses may elevate the risk of kidney stones. Additionally, both standalone vitamin D supplementation and combined vitamin D and calcium supplementation have been shown to increase serum 25(OH)D levels. The simultaneous administration of calcium and vitamin D has been linked to enhanced bone mineral density (BMD) [66]. Notably, both vitamin D and calcium deficiency are acknowledged as concurrent bone risk factors, given their indispensable roles in bone health [67,68]. Specifically, vitamin D contributes to calcium absorption, thereby significantly contributing to bone health [69]. Understanding these intricate relationships is vital for formulating effective strategies to maintain optimal bone health across different life stages.

### 1.7. Pregnancy and Breastfeeding

During pregnancy, a woman’s body undergoes significant changes, driven by the crucial task of nourishing the fetus through the placenta over a period of nine months. These physiological changes can include elevated blood pressure [70]. Pregnancy-induced hypertension affects one in ten pregnancies and stands as a leading global cause of maternal mortality [36,37,71]. Research has consistently demonstrated that vitamin D deficiency during pregnancy is linked to adverse outcomes [72], including an increased risk of preeclampsia [73,74,75,76,77], gestational diabetes mellitus (GDM), preterm birth, births of small-for-gestational-age (SGA) infants, and weakened fetal bones [4]. Moreover, it can impact the health of both newborns and later children.

Numerous studies have highlighted the potential benefits of vitamin D supplementation during pregnancy, leading to increased concentrations of 25(OH)D in the blood and a reduced risk of pregnancy complications. Specifically, a higher dose of vitamin D, beginning at 1000 IU daily, is recommended, especially for non-Caucasian races, high-risk women with a BMI > 30 kg/m^2^, those residing in high latitudes, or those delivering between November and March [78]. It is imperative to establish comprehensive guidelines for supplement use during pregnancy to ensure optimal maternal and fetal health.

Even during the breastfeeding period, a woman’s micronutrient needs, including vitamin D, do not return to pre-pregnancy levels. Studies have indicated that maternal vitamin D intake ranging from 4000 to 6400 IU per day can effectively fulfill the infant’s vitamin D requirements, particularly for mothers who exclusively breastfeed [79,80]. Addressing these nutritional needs is crucial for the continued well-being of both the mother and the breastfeeding infant.

### 1.8. Vitamin D, Immune System, and Other Effects

Vitamin D receptors are distributed throughout the body, including immune cells, which exert a direct influence on immune system health [81]. The primary condition directly associated with vitamin D deficiency is tuberculosis. It has been reported that sunlight’s ultraviolet radiation can positively impact tuberculosis treatment. However, studies have found no statistically significant effects with vitamin D supplementation [82,83].

Low vitamin D levels have also been linked to respiratory tract infections. Specifically, an inverse correlation was identified between the risk of upper respiratory diseases and levels of vitamin D (25(OH)D) in cord blood. This correlation led to infections in infants during the first 3 months of life and wheezing at 15 months [84]. Neonates born with vitamin D levels less than 20 ng/mL exhibited a six-fold higher risk of respiratory syncytial virus bronchiolitis at 1 year of age compared to those born with levels > 30 ng/mL [85]. Vitamin D supplementation, especially in severe deficiencies, has proven vital in acute respiratory infections. In asthma, it was reported that vitamin D supplements during pregnancy may reduce the risk of wheezing episodes in children. Notably, low 25(OH)D3 levels are directly related to asthma [86,87].

Another disease associated with vitamin D deficiency is atopic dermatitis, with affected patients exhibiting lower than normal serum vitamin D levels [88]. One study suggested that vitamin D supplementation might benefit children with winter-related dermatitis [89]. Calcitriol inhibits dendritic cell maturation, reducing acquired immunity activation and increasing the risk of autoimmune diseases such as diabetes mellitus, multiple sclerosis, and inflammatory bowel disease. Additionally, despite insufficient and inconclusive data, vitamin D deficiency may be linked to a higher incidence of cancer, musculoskeletal pain, migraines, and psychiatric disorders such as schizophrenia, dementia, and depression [90,91]. Further research is necessary to comprehensively elucidate these associations.

### 1.9. Vitamin D and Diabetes Mellitus

During pregnancy, the woman’s body undergoes various changes, including alterations in vitamin D metabolism. Vitamin D deficiency has been linked to adverse pregnancy outcomes, notably preeclampsia and diabetes mellitus [92,93,94,95]. Diabetes mellitus is recognized as an independent risk factor for preeclampsia [61]. Studies have indicated that insulin resistance might contribute significantly to the pathophysiology of preeclampsia [95]. Evidence from comparisons between normally healthy pregnant women at lower risk for preeclampsia and those with preeclampsia revealed that these women were more likely to develop insulin resistance and diabetes mellitus both during pregnancy and in subsequent years [96,97,98]. Notably, insulin resistance identified at 22–26 weeks of gestation independently predicted preeclampsia, establishing diabetes as an autonomous risk factor [97]. Additionally, a reciprocal relationship between preeclampsia and diabetes mellitus has been observed, with preeclampsia being associated with an elevated risk of developing diabetes mellitus [99,100,101,102,103]. Pregnant women with type 1 diabetes and preeclampsia exhibit lower vitamin D levels compared to women without preeclampsia [61,104,105,106].

The objective of this review is to assess whether the concentration of vitamin D in the serum of pregnant women, in all its forms, is correlated with the risk of preeclampsia during pregnancy. This evaluation explores whether vitamin D levels can function as both protective and risk factors or even prognostic indicators of the disease. Additionally, the present review aims to summarize the effects of vitamin D supplementation during pregnancy to evaluate its potential as a preventive measure against preeclampsia.

## 2. Materials and Methods

For this literature review, we searched Scopus and PubMed using the words “Vitamin D” and “preeclampsia” to retrieve articles related to vitamin D supplementation or vitamin D serum levels in women with preeclampsia until May 2023. We also used the Boolean Operator NOT with the word “Review” in Scopus and the words “Review”, “Systematic Review”, and “Meta-Analysis” in PubMed. No language restrictions were applied, and all types of studies, excluding animal studies, were included in the review.

## 3. Results

By using two different search strategies, we identified 261 and 36 studies in Scopus and PubMed, respectively. We excluded duplicates and articles whose full text is not available or not relevant to our review. Finally, 31 studies that met the inclusion criteria were included in this review (Figure 1).

### 3.1. Vitamin D Supplementation

In total, 10 out of the 31 studies are randomized clinical trials investigating the role of vitamin D supplementation in preeclampsia (Table 1) [34,107,108,109,110,111,112,113,114,115]. Participants’ (n = 4614) mean age varied, ranging from 18 to 40 years. Studies were conducted in Saudi Arabia, Iran, Ukraine, Romania, Finland, and the USA. Supplementation dose varied among the studies, ranging between 400 IU and 300,000 IU daily.

All the included studies demonstrated that vitamin D levels are related to preeclampsia, except one study, which not found association between vitamin D and preeclampsia [109]. The vitamin D supplementation dose plays a crucial role in the disease, as the higher the dose, the lower the risk of developing preeclampsia.

### 3.2. Observational Studies

Nine studies were case–control studies [14,18,31,32,45,55,73,116,117] (Table 2). The studies mainly contained case–control groups, but there was, for example, a nested case–control group, etc. A case–control study conducted in Pennsylvania tried to find out if there is an association between vitamin D levels and preeclampsia in 274 women aged between 14 and 44 [73]. This study showed that there is an association between vitamin D levels and preeclampsia [73]. Similar to this result, Anderson et al. 2015 showed that there is a significant association between vitamin D and preeclampsia [116]. Pashapour et al. in a 2019 case–control study based on 160 pregnant women (age range 24–35) conducted in Iran and measuring the levels of 25(OH)D in both cases and control groups found an association between vitamin levels and preeclampsia [117]. Preeclamptic women tend to be older with a higher body mass index (BMI) [117]. Serrano et al. (2018) conducted a study in Colombia on a large number of pregnant women and analyzed sociodemographic data and maternal serum vitamin D levels, finding that 52% (significant percentage) of those with preeclampsia were vitamin D-deficient [32]. Benachi et al. (2019) conducted a nested case–control study in France and Belgium among a large sample of 402 total women. In this study, the women were divided into two groups, one of which consisted of the controls. Vitamin D (as 25(OH)D) was measured during the first and third trimester. The aim was to determine the association between vitamin D and preeclampsia during the first trimester of pregnancy. It turned out that there was no significant association between preeclampsia and vitamin D in this trimester of pregnancy. Still, women with vitamin D sufficiency in their last trimester of pregnancy had a lower risk of preeclampsia [45]. The studies from Yuan et al. 2021 and Haile et al. (both 2021) almost showed the same result (a significant association between maternal vitamin D and preeclampsia) [31,55]. The former was conducted in China, analyzing serum 25(OH)D concentrations in the second and third trimesters and achieving the aforementioned result [55]. The total number of participants was 610 (488 controls and 122 cases) [55]. Women with preeclampsia tended to be older and have higher BMI scores, and maternal vitamin D levels were lower in women with preeclampsia [55]. The number of participants was approximately half of the aforementioned study, and the methodology included taking maternal blood samples from both cases and controls [31]. A matched case–control study conducted in China from March 2016 to June 2019 involved a total of 1180 unevenly distributed participants; in this study measuring serum concentrations of vitamin D in its various forms demonstrated that dietary intake was negatively associated with preeclampsia risk [18] while Forde et al. (2021) found no association [118].

Seven of the included studies were cohort studies [30,43,60,72,119,120,121] (Table 2). In 2016, a cohort study by Bärebring in Sweden involving a total of 2000 women with an average age of 31.3 years was conducted [60]. Only 4% of the total participants developed preeclampsia; thus, it seems that it is not related to the hypertension caused during pregnancy [60]. The same result was obtained by Chia et al. (2018) in a GUSTO study in Singapore, with the only difference being that there no blood samples were taken and that the result was obtained only by analyzing dietary intake and quality [121]. In the United Kingdom, Tamblyn et al. (2017) conducted a study involving 88 pregnant women divided into four groups, and no significant difference regarding age was found [43]. The results showed a significant association between vitamin D levels and preeclampsia. In a large study involving 3703 participants from 12 states of the US, the authors of [119] divided subjects into two groups and, in both, measured [25(OH)D] < 26 w, showing that vitamin D deficiency may be a risk factor for preeclampsia. Another cohort study conducted in Australia included Vitamin D metabolite data at 28–32 w and birth, and Treg ratios in cord blood showed that higher maternal 25(OH)D3 levels were associated with increased free Treg and aTreg [120]. Also, there is a positive correlation between cord blood 25(OH)D3 levels and cord blood aTreg, and it has been reported that 25(OH)D3 may play a role in neonatal immunity [120]. Generally higher maternal levels of 25(OH)D3 increase free Treg cells such as aTreg cells. A positive correlation was found between 25(OH)D3 concentration in cord blood serum and αTregs in cord blood serum. Additionally, 25(OH)D3 concentration may be correlated with infant immunity [120]. In one study conducted in France from June 2008 to October 2010 [30], maternal blood 25(OH)D concentration was measured at five different times during pregnancy, 20, 24, 28, 32, and 36 weeks. The results showed that 43% of participants who developed preeclampsia had insufficient vitamin D levels during pregnancy, so there was a strong negative association between vitamin D levels at 32 w and the risk of developing preeclampsia [30]. Boyle et al. (2016) conducted a cohort study on a total number of 1710 women in Australia. In the vitamin D-replete pregnancy cohort, 25-hydroxyvitamin D concentration did not predict pregnancy outcomes, including preeclampsia [72]. Burris et al.’s 2014 cohort study was conducted in the USA and involved 1591 women; they measured the levels of 25(OH)D at 16.4–36.9 w of pregnancy and came to the conclusion that no frequency was detected on average at 27.9 weeks of pregnancy. It was also found that for every 25 nmol/L, the risk of preeclampsia increased by 33% [122]. Kiely et al. (2016) conducted a cohort study in Ireland, and by measuring the levels of three forms of vitamin D during the 15th week of pregnancy, it was observed that levels >75 nmol/L were protective with respect to preeclampsia [123].

### 3.3. Effect of a Low-Glycemic-Index Diet on Vitamin D

One study evaluated the effect of a low-glycemic-index diet on vitamin D levels [118]. In this observational study, which took place in Dublin between 2007 and 2011, 415 women were involved and followed up on for 5 years (Table 2). The mean maternal age was 32.1 years. No significant difference was observed between the dietary intake of vitamin D and overall maternal diet with the risk of preeclampsia [118].

**Table 2 diseases-11-00158-t002:** Studies monitoring the effects of intaking various vitamin D concentrations on lifestyle.

Authors	Country	Study Design	Maternal Age	N Total	Duration of Investigation	Eligibility Criteria or Exclusion Criteria	Methodology of the Study	N/Group	Mean Age/Group	Preeclampsia	Statistically Significant Correlation between Preeclampsia and Vitamin D? (Yes/No)
Anderson, 2015 [116]	USA, Dakota	Case–control study	24–26	48	n/a	Eligibility: Nulliparous women, >18 years (<14 w of pregnancy)	Dietary intake- supplementation of vitamin D in the previous 3 months. Normotensive and PE groups determined by blood pressure and proteinuria. RIA to adjust the maternal [25(OH)D] serum.	Normotensive: n = 37 Gestational hypertension: n = 11	Normotensive: 24.2 ± 0.62 Gestational hypertension: 25.3± 0.72	There is reference to SBP/DBP and hypertension during pregnancy	Yes
Bärebring, 2016 [60]	Sweden	Cohort study	31.3 (mean)	2000	Fall 2013 (2 September–8 November) Spring 2014 (24 February–13 June)	Exclusion: pregnancy exceeding 16 w gestation, miscarried before 20 w, lost to follow-up	Blood samples <16 w (8–12) and after 31 w (32–35), questionnaire for lifestyle. After delivery: BP, proteinuria, preexisting medical conditions, weight, height, etc. Available samples T1, T3 (N = 1827)	n/a	n/a	4%	No
Benachi, 2020 [45]	Belgium and France	Nested case–control study/prospective observational cohort	n/a	402	April 2012–February 2015	Patients were excluded from selection of controls if we could not be sure whether preeclampsia occurred or not (patients not followed up to delivery or no data on blood pressure or proteinuria), if pregnancy was interrupted (abortion, intrauterine fetal death) or if there were no data on delivery. The sample of eligible controls was obtained from controls without preterm delivery (37 WA), whose newborn was alive in the delivery room and presented no intrauterine growth restriction <5th percentile) at birth, with vitamin D measurement available in both the first and third trimesters and with no missing data on any matching factors.	Vitamin D insufficiency in the first trimester and preeclampsia later in pregnancy. A bolus vitamin D dose (100,000 IU of cholecalciferol) was prescribed to the patients at the seventh month of pregnancy according to current French recommendations. The main outcome measure was serum 25(OH)D status in the first trimester.	Cases n = 93 controls n = 319	cases 32.2 ± 5.9, controls 31.7 ± 5.0	No percentage per group	No
Robinson, 2010 [14]	USA	Case–control study	26 (24 + 28) (mean)	150	n/a	Eligibility criteria for EOSPE: American College of Obstetrics and Gynecology and diagnosis prior to 34 w Exclusion for EOSPE: chronic hypertension, pregestational diabetes, renal disease, lupus, multiple gestation Exclusion criteria for controls: EOSPE criteria	Demographic (BMI, gestational age, maternal, SBP, DBP, urine protein) and outcome data of plasma 25(OH)D analysis	Cases: n = 50 Controls: n = 100	Cases: 24 (21–30) Controls: 28 (23–32)	Cases: 42% Controls: 10%	Yes
Tamblyn, 2017 [43]	United Kingdom	Cross sectional analysis	n/a	88	n/a	n/a	Placental biopsies, analysis of DBP, albumin and free vitamin D metabolites, blood samples, demographic characteristics	First trimester—healthy: n = 25 third trimester—healthy: n = 21 PET: n = 22 Non pregnant: n = 20	No significant differences between ages	n/a	Yes
Pashapour, 2019 [117]	Iran, West Azerbaijan	Case–control study	24–35	160	January–May 2016	Inclusion criteria: singleton pregnancy, no medical disorders (diabetes, kidney, hypertension), no history of vitamin intake during pregnancy, no smoking, no BMI > 30 kg/m^2^	Measurement od 25(OH)D in pregnant women with and without preeclampsia	Preeclampsia group: n = 80 Healthy group: n = 80	Preeclampsia group: 28.83 ± 6.7 Healthy group: 24.13 ± 8.4	Preeclamptic women tend to be older, >BMI, 30–35 years old more prone to get PE. Lower vitamin D increased PE	Yes
Forde, 2021 [118]	Dublin	Observational study	32.1 (mean)	415	2007–2011 (5-year follow up)	Exclusion criteria: history of gestational diabetes, concurrent medication, age < 18, multiple pregnancy	Follow up women in low-glycemic-index diet, maternal concentration of vitamin D and mother’s age	The is no separation into subgroups	32.1	No correlation between vitamin D consumption and blood pressure during first–third trimester and 5 y follow up	No
Huang, 2022 [18]	China	Matched control study	n/a	1180	March 2016–June 2019	Cases inclusion criteria: diagnosed with preeclampsia according to the guidelines for hypertensive disorders in pregnancy Controls criteria: no preeclampsia	Serum concentrations of 25(OH)D2 and 25(OH)D3	Cases: n = 532 Controls: n = 648	Cases: 30.88 (mean) Controls: 31.03 (mean)	Dietary intake was negatively associated with preeclampsia risk. Higher vitamin d intake or serum concentration are associated with lower risk of preeclampsia	Yes
Boyle, 2016 [72]	Australia	Cohort study	30.3 (mean)	1710	2005–2008	n/a	Non-fasting serum samples were collected at 15 weeks of gestation. Women were screened for GDM between 24 and 28 weeks of gestation with a non-fasting 50 g polycose challenge in community laboratories, according to the Auckland District Healt Board Guidelines	Cases: n = 93, controls n = 319	cases 32.2 ± 5.9, controls 31.7 ± 5.0	Vitamin D did not predict preeclampsia	No
Raia-Barjat, 2021 [30]	France, Nimes	Ad hoc study of a previous cohort	32 ± 5 (mean)	200	June 2008–October 2010	Inclusion criteria: High risk for preeclampsia that included diabetes, chronic hypertension, systemic lupus, antiphospholipid syndrome, history of cardiovascular disease, etc. Exclusion criteria: twin pregnancy, history of fetal death, chromosomal IUGR, venous thromboembolism	Monitoring blood samples for analysis of serum 25(OH)D at 20, 24, 28, 32, 36 w	PMC: n = 43 No PMC: n = 139	PMC: 32 ± 5.3 No PMC: 32.1 ± 5.1	43% developed PMC. Vitamin D levels at 32 w were lower between patients with PMC. Strong relationship between vitamin D levels and the risk of PMC at 32 w	Yes
Yuan, 2021 [55]	China	Nested case–control study	29–34	610	March 2012–February 2015	Exclusion criteria: loss of basic information, medical abortion, multiple pregnancy, and the use of assisted reproductive technology or 25(OH)D concentrations beyond the assay detection limit	Serum 25(OH)D was at first examination in second or third trimester. Diagnosis of preeclampsia during other examination	Cases: n = 122 Controls: n = 488	Cases: 30.6 ± 3.7 Controls: 29.6 ± 2.8	Women with preeclampsia were older, >BMI. Maternal vitamin D levels were lower in women with preeclampsia	Yes
Serrano, 2013 [32]	Colombia	Case–control study	<25 years except HELLP syndrome	2028	December 2000–February 2012	Inclusion criteria for cases: BP ≥ 140/90 mmHg, 24 h proteinuria ≥ 300 mg or ≥2+ in urine after 20th week Exclusion criteria: history of hypertension, diabetes mellitus, renal or autoimmune diseases	Sociodemographic data for all participants, samples of maternal 25(OH)D and quantification	Cases: n = 1013 Controls: n = 1015	Cases: 19.1 Controls: 18.7	52% of women with preeclampsia were vitamin D deficient.	Yes
Bodnar, 2007 [73]	Pennsylvania	Prospective cohort study	14–44	274	1997–2001	Inclusion criteria: <16 w gestation, singleton pregnancy, 14–44 y.o., planning deliver at Magee- Women’s Hospital	Quantitation of serum 25(OH)D, examination of predisposing factors for preeclampsia	Cases: n = 55 Controls: n = 219	Cases: <20 y 20, 20–29 y 52.7, >30 y 27.3 Controls: <20 y 37.4, 20–29 y 47.5, >30 y 15.1	Serum [25(OH)D] were lower 15% in cases	Yes
Haile, 2021 [31]	Ethiopia	Unmatched case–control	<20, 20-34, ≥35	344	n/a	n/a	Blood samples from both cases and controls	Cases: n = 86 Controls: n = 258	Cases: <20 y 9, 20–34 y 64, ≥35 y 13 Controls: <20 y 26, 20–34 y 200, ≥35 y 32	n/a	Yes
Chia, 2018 [121]	Singapore	GUSTO study	18–50	1051	June 2009–September 2010	Exclusion criteria: Type 1 diabetes mellitus, chemotherapy, psychotropic drugs, parents or spouses had different ethnicities	Dietary intakes and analyze diet quality at 26–28 w of pregnancy. Infant outcomes	No separation into groups	n/a	Maternal diet quality does not have any relationship with SGA or prematurity	No
Bodnar, 2014 [119]	USA (12 States)	Case–cohort study	n/a	3703	1959–1966	Exclusion criteria: preexisting conditions, entry to care at ≥26 w	In both cases and controls, measurement of [25(OH)D] at ≤26 w	Subcohort: n = 3068 Cases: n = 717	Subcohort: <20 23%, 20–29 61%, ≥30 16% Cases: <20 37%, 20–29 43%, ≥30 20%	Mothers who developed preeclampsia tended to be black, nulliparous, <20/ ≥ 30 years old, smokers, less educated, overweight, etc. There is no significant association between maternal [25(OH)D] and preeclampsia. Although it may be a risk factor	Yes
Thorsen, 2021 [120]	Australia	Cohort study	n/a	1074 pregnant women- and 1074 infants	2010–2013	Exclusion criteria: birth <32 w, serious disease, genetic abnormality	Vitamin D metabolite data at 28–32 w and birth, and Treg proportions in CB	No separation into groups	n/a	Higher maternal 25(OH)D3 levels were associated with increased free Treg, and aTreg. A positive correlation was found between CB 25(OH)D3 levels and CB aTreg. 25(OH)D3 may play a role in neonatal immunity	Yes
Burris, 2014 [122]	USA	Prospective prenatal cohort study	32.1 (mean)	1591	n/a	n/a	Among women enrolled in the Project Viva prenatal cohort in Massachusetts, we examined associations of 25(OH)D levels obtained at 16.4–36.9 weeks of gestation (mean 27.9 weeks) with hypertensive disorders of pregnancy, including preeclampsia (56/1591, 3.5%) and gestational hypertension (109/1591, 6.9%)	category per plasma [25(OH)D] (nmol/L): <25 n = 81, 25-<50 n = 472, 50-<75 n = 743, ≥75 n = 295	3.8%	No, it was not detected an association at mean 27.9 w gestation and preeclampsia	No
Kiely, 2016 [123]	Ireland	Cohort study	n/a	1768	March 2008–January 2011	n./a	Non-fasting serum samples were collected at 15 weeks of gestation. Women were screened for GDM between 24 and 28 weeks of gestation with a non-fasting 50 g polycose challenge in community laboratories, according to the Auckland District Health Board Guidelines.	cases n = 93, controls n = 319	3.8%	Yes, it was protective at levels >75 nmol/L	Yes

### 3.4. Vitamin D Levels and Preeclampsia Incidence in Case of Comorbidities

We also found a population-based study of women with comorbid pre-existing diabetes mellitus conducted in three countries (Table 3): Norway, Australia, and the USA [61]. There was no mention of the maternal ages of the pregnant women who participated into the study, but the total number of participants was 175. Participants assigned to the healthy group had the same characteristics (no hypertension, no proteinuria, and no microalbinuria at enrollment). At the same time, the characteristics of the preeclamptic women were new-onset hypertension and proteinuria at gestational week >20. There were 23 total diabetes mellitus cases among women with preeclampsia (mean age: 28.5± 5.6), 24 normotensive women with diabetes mellitus (mean age: 29.9± 5.6), and 19 normotensive women without diabetes (mean age: 31.4± 4.5). Regarding preeclampsia, vitamin D deficiency was shown to be more common in women with type 1 diabetes, but no maternal concentration predicted preeclampsia. At the second and third visit, [25(OH)2D] was associated with preeclampsia in women with type 1 diabetes, as were lower Vitamin D Binding Protein (VDBP) concentrations. In general, women who had diabetes mellitus had an increased risk of vitamin D deficiency, and this study found an association between vitamin D levels and preeclampsia [61]. A further analysis of the VDAART study was carried out by Mirzakhani et al. in three different American states in 2018. The eligibility criteria for participation in the study included a personal history of diagnosed asthma or allergies, respectively, or a personal history of diagnosed asthma or allergies, respectively, on the partner’s side [87]. Women were excluded from the study if they were smokers or used other nicotine products, had any chronic conditions, had experienced multiple pregnancies, had achieved pregnancy via assisted reproduction techniques, had parathyroid and thyroid disease, had kidney stones and sarcoidosis, had an intake of over 2000 IU/day of vitamin D supplements, or had taken illicit drugs in the last six months [87]. In the study, the women were divided into two groups: asthmatic (n = 327) and non-asthmatic (n = 489). Preeclampsia presented in 29 asthmatic and 39 non-asthmatic women. A significant difference between pregnant women with uncontrolled asthma and non-asthmatic women or women with controlled asthma was reported, but no significant difference was found between pregnant women with controlled asthma and non-asthmatic women. Thus, taking vitamin D supplements appeared to reduce the risk of preeclampsia, which occurred due to asthmatic status control [87].

### 3.5. Studies Showing No Association between Vitamin D Levels and Preeclampsia

Overall, among the 31 studies, only 7 of them found no association between vitamin D and preeclampsia. More specifically (Table 2), in a cohort study, Bärebring et al. (2016) [60] took blood samples in the first and third trimester to measure 25(OH)D concentration and found that in the first trimester, there was no association between 25(OH)D concentration and the incidence of preeclampsia, since preeclampsia occurred in only 4% of the total sample population. Another study that aimed to determine whether there was an association between preeclampsia and a low-glycemic-index diet, vitamin D consumption and 25(OH)D status found no association, something that was confirmed during the next 5 years of follow up [118]. Similar to the aforementioned, the study by Benachi et al. (2020), which involved participants from Belgium and France and lasted for three years, did not find any correlation between vitamin D and preeclampsia [45]. A cohort study conducted in Australia between 2005 and 2008 also found no correlation, as vitamin D levels did not predict preeclampsia [72]. A GUSTO cohort study [121] monitored the diet of 1051 women over days and found no association between maternal diet, prematurity, and other adverse pregnancy outcomes. In addition, Naghshineh et al. (2016) (Table 1), in a study that included an intervention (placebo versus vitamin D supplementation [109]) found no association between vitamin D concentration during pregnancy and preeclampsia.

## 4. Discussion

A total of 31 different studies were included in this review to summarize recently published information on vitamin D levels and preeclampsia risk. Different forms of research papers were included in our review. Through this article, we aimed to better understand the importance of vitamin D levels in relation to the development of preeclampsia in pregnant women regardless of origin and pregnant women with comorbidities such as diabetes mellitus type 1 [61], as well as the effect of vitamin D supplementation. The worldwide prevalence of preeclampsia is between 2 and 8%, with its most severe effect being maternal death [18,30,34,35,36,37,38,71]. In these cases, the fetus either fails to come into the world or the infant is born but experiences adverse effects later in life. Various types of interventions and monitoring procedures for the pregnant women who participated in the research studies were carried out, with most studies involving the administration of vitamin D supplements [34,107,108,109,110,111,112,113,114]. Results derived from supplementing pregnant women with vitamin D, in relation to the course of preeclampsia, are encouraging [34,107,108,110,111,112,113,114]. In [73], in many cases, vitamin D deficiency or insufficiency was an independent risk factor, and there was no need to consider additional co-factors to calculate the results, as the pregnant women were otherwise healthy. In the studies that reported the minimum normal level of vitamin D, we found the following characterizations of vitamin D levels: deficiency at levels <20 ng/mL and insufficiency at levels <32 ng/mL [61]. It is mentioned, as our study has also showed, that the maintenance of normal blood pressure during pregnancy is something that affects both the mother and the baby she is carrying.

Research has shown that an adequate amount of vitamin D (25(OH)D concentration levels ≥ 32 ng/mL) is itself a protective factor [27,117]. It should be noted that levels of deficiency can be divided into three individual levels [14]: severe deficiency (blood serum vitamin D concentration of <10 ng/mL), moderate deficiency (serum vitamin D concentration of 10 to 20 ng/mL), and simple deficiency (reportedly >20 ng/mL but obviously below adequate levels). It can be realized that there is indeed an association between the concentration of vitamin D during pregnancy and the risk of preeclampsia. In pregnancy, as in breastfeeding, the pregnant woman’s need for vitamin D increases [79,80]. Therefore, in such cases, the situation is aggravated by a possible deficiency of the vitamin D. [124]. Thus, the outcome of pregnancy is inextricably linked to blood pressure levels, as shown in [125,126]. As mentioned at the beginning of this paper, preeclampsia can be a disease of the placenta, and the mechanism of the disease is unknown. It has been suggested that vitamin D affects the early placenta and plays an important role as an immunomodulator in the mother–fetal interaction [127]. One study proved that low levels of vitamin D (not a rare phenomenon) can affect the development of the placenta. In addition, it is worth mentioning that as the body mass index increases, so does the risk of vitamin D deficiency [127].

In some studies, comorbidities suffered by pregnant women appeared to play an important role. For example, diabetic women have a higher risk of developing preeclampsia anyway, and so it is possible that supplemental oral vitamin D, in this particular population, may be protective against the disease or even prevent the possibility of experiencing it. Thus, the health of mothers could theoretically improve with even more frequent monitoring of their vitamin D levels during pregnancy. Further improvements could be made if their dietary vitamin D intake was simultaneously monitored to ensure that they obtain the highest possible levels of vitamin D from their diet, although the dietary sources of vitamin D remain scarce.

Further research should be performed to explore the effect of vitamin levels per trimester to find out if the deficiency is detected early, even in the first weeks of pregnancy; it may be possible to restore it to normal levels in a short time to ensure better pregnancy outcomes. However, this requires a large sample and a sufficient number of surveys, and research activities should not be limited to certain races or groups of people. Another important suggestion for further research could be the exclusive study of populations that have had any history of preeclampsia manifestation for several generations in any (if it is more than one pregnancy per individual) pregnancy. This may facilitate a better understanding of exactly where the problem is located since we are talking about clearly healthy groups without comorbidities, which, over the years, have been documented to burden the specific disease, and the exclusive eligibility factor of women is low vitamin D levels. If we assume that it is certain that vitamin D at a low or marginally low level during pregnancy determines the later manifestation of either late- or early-preeclampsia, we could further explore other vitamins, minerals, or trace elements that accelerate and improve its absorption by the body. This could enable us, even with the use of supplements, to improve their absorption and therefore achieve a sufficient amount more quickly, preventing fatal consequences for the mother and the fetus.

There are different forms of vitamin D, and vitamin D goes through different stages in our body. It would be good to carry out a thorough investigation wherein pregnant women with same-stage preeclampsia and at the same week of pregnancy, ensuring that measurements are made uniformly and considering the same number of fetuses in each pregnancy. In this way, it would be possible to identify the factor that can be used as a more reliable prognostic indicator to predict the risk of preeclampsia with high certainty. Something important that deserves further investigation is the rate of vitamin D absorption by age through the sun, diet, and dietary supplements. We believe it would be better to exclude populations of women with a history of maternal vitamin deficiency and limit research scopes to a very specific age group. These potential studies could involve three groups of women: women who mainly intake vitamin D via their diet, women whose main source of vitamin D is the sun, and women with a previous deficiency which was corrected via dietary supplement administration whose main sources of vitamin D are supplements.

## 5. Conclusions

Out of the 31 articles included in our review, only 7 of them (negligible rate—22.6%) showed no significant difference between vitamin D levels and preeclampsia regardless of comorbidity. It should be noted that low vitamin D levels may have an effect on newborns and children, in addition to the immediate health risks posed to the mother during or after childbirth, including death. In conclusion, most studies showed an association between maternal vitamin D levels during pregnancy and preeclampsia. It is important to understand the pathophysiology of this particular disease, as despite the fact that it has been documented in studies, it is a complex disease, and perhaps for this reason, it is not fully understood. In many studies, however, supplementation-related results were encouraging, as pregnant women reached desirable and protective levels of vitamin D, avoiding adverse pregnancy outcomes.

## Figures and Tables

**Figure 1 diseases-11-00158-f001:**
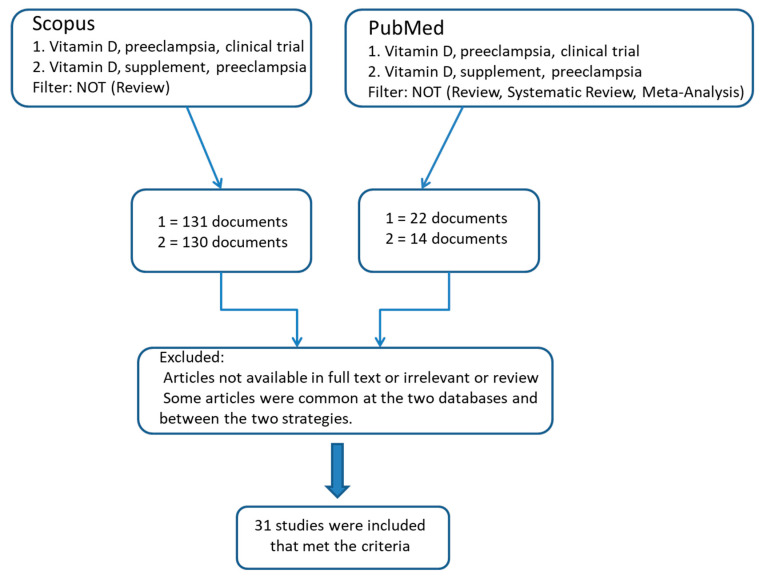
Flow chart diagram of the included studies.

**Table 1 diseases-11-00158-t001:** Results based on vitamin D supplementation.

Authors	Country	Study Design	Maternal Age	N Total	Duration of Investigation	Intervention-Groups	N/Group	Mean Age/Group	Preeclampsia	Statistically Significant Correlation between Preeclampsia and Vitamin D? (Yes/No)
Ali, 2019 [107]	Saudi Arabia	Randomized controlled study	20–40	164	October 2012– October 2015	Group 1: 400 IU vitamin D3; group 2: 4000 IU vitamin D3	Group 1: n = 81 Group 2: n = 83	Group 1: 29.3 Group 2: 29.4	Group 1: 7.4% Group 2: 1.2%	Yes
Behjat Sasan, 2017 [34]	Iran	Randomized controlled clinical trial	24–35	142	n/a	Group 1: 50,000 IU vitamin D3/2 weeks; group 2: placebo	Group 1: n = 70 Group 2: n = 72	Group 1: 32.04 ± 5.901 Group 2: 29.77 ± 5.21	Total: 23.2% Group 1: 15.7% Group 2: 30.6%	Yes
Manasova, 2021 [108]	Ukraine	Randomized controlled clinical trial	27–33	54	2017–2020	Group 1: multivitamin mineral complex (including cholecalciferol 500 IU); group 2: multivitamin mineral complex until 16 w and then 2000 IU until the end of the pregnancy	Group 1: n = 25 Group 2: n = 29	Group 1: 27.4 ± 4.4 Group 2: 28.2 ± 4.6	n/a	Yes
Naghshineh, 2016 [109]	Iran	Double-blind randomized controlled trial	25 ± 4.1 (mean)	140	May–January 2012	Group 1: 600 IU vitamin D/day; group 2: placebo	Group 1: n = 68 Group 2: n = 70	Group 1: 25 ± 3.8 Group 2: 24.8 ± 4.4	Group 1: 2 women Group 2: 7 women	No
Dahma, 2022 [110]	Romania	Single-centric case–control	25–34	198	2018–2022	Group 1: No supplementation; group 2: Low dose, 2000 IU during first trimester; group 3: High dose, 4000 IU	Group 1: n = 59 Group 2: n = 63 Group 3: n = 76	n/a	Group 1: 18.6% Group 2: 9.5% Group 3: 5.3%	Yes
Rostami, 2018 [111]	Iran	Stratified randomized field trial	18–40	Phase 1: 2500 Phase 2: 800	n/a	Moderate-deficiency: Group 1: 50,000 IU D3/w for 6 w; group 2: same as Group 1 + 50,000 IU D3/month until delivery; group 3: single dose of 300,000 IU D3; group 4: same as Group 3 + 50,000 IU/month until delivery. Severe-deficiency: Group 5: 50,000 IU/w for 12 w; group 6: same as Group 5 + 50,000 IU/month until delivery; group 7: two doses of 300,000 IU for 6 w; group 8: same as Group 7 + 50,000/month until delivery	n/a	Screening site: 25–32 (mean 29) Non-screening site: 25–32 (mean 29)	Moderate-deficiency screening site: 7% Severe-deficiency screening site: 8% Total screening site: 8% Moderate-deficiency non-screening site: 13% Severe-deficiency-non-screening site: 23% Total non-screening site: 17%	Yes
Xiaomang, 2021 [112]	Finland	Open label randomized study	20–40	450	January 2016– December 2018	Group 1: low dose 400 IU/d; group 2: middle dose 1500 IU/d; group 3: high dose 4000 IU/d	Group 1: n = 135 Group 2: n = 134 Group 3: n = 138	Group 1: 27.76 ± 3.16 Group 2: 28.54 ± 3.27 Group 3: 28.94 ± 3.21	Group 1: 9.63% Group 2: 6.72% Group 3: 1.45%	Yes
Mirzakhani, 2016 [113]	Boston Massachusetts, Missouri, California	VDAART study	18–40	881, available data for 816	October 2009– July 2011	Prenatal vitamins 400 IU cholecalciferol G1:4400 IU/day; G2: placebo/day	G1: n = 408 G2: n = 408	Group 1: 27.5 Group 2: 27.2	For 8.2% of all subjects, supplementation at week 10–18 did not reduce preeclampsia. Sufficient vitamin during enrollment and pregnancy lower the risk of preeclampsia	Yes
Samimi, 2016 [114]	Iran	Randomized double-blind placebo-controlled clinical trial	18–40	60	September 2014–February 2015	Group 1: 50,000 IU vitamin D3; Group 2: placebo	Group 1: 50,000 IU vitamin D3 n = 30: Group 2: placebo n = 30	Group 2: n = 27.1, Group 1: n = 27.3	Vitamin D+ Ca for 12 w affected blood pressure	Yes
Azami, 2017 [115]	Iran	Randomized controlled clinical trial	31.63 ± 6.13 (mean)	90	n/a	Group 1 received Ferrous sulfate (1 tablet/day) + one tablet of Claci-care multimineral-vitamin D containing 800 mg calcium, 200 mg magnesium, 8 mg zinc and 400 IU Vitamin D3 per day, group 2 received Ferrous sulfate (1 tablet/day) + 250 mg vitamin C + 55 mg vitamin E, and group 3 (controls) received only one Ferrous sulfate tablet daily	Three equal (n = 30) groups	Group 1: 33.03 ± 6.49, Group 2: 31.73 ± 6.41, Group 3 (control): 30.15 ± 5.28	n/a percentage per group	Yes

IU: International Units; VDAART: Vitamin D Antenatal Asthma Reduction Trial.

**Table 3 diseases-11-00158-t003:** Vitamin D levels and preeclampsia incidence in cases of comorbidities.

Authors	Country	Study Design	Maternal Age	N total	Duration of Study	Eligibility Criteria or Exclusion Criteria	Methodology of the Study	N/Group	Mean Age/Group	Preeclampsia	Statistically Significant Correlation between Preeclampsia and Vitamin D? (Yes/No)
Kelly, 2020	Norway, Australia, USA	Cohort study	n/a	66	n/a	Eligibility criteria for health: hypertension free, no proteinuria, no microalbuminuria at enrollment. Eligibility criteria for preeclampsia: new- onset hypertension and proteinuria at >20 w	Total, bioavailable, free 25(OH)D, 1,25(OH)2D and VDBP at week 12, 22 and 32	Type 1 diabetes mellitus subgroups: Diabetes mellitus with preeclampsia: n = 23. Normotensive with diabetes: n = 24. Normotensive and non-diabetic as controls: n = 19.	Diabetes mellitus with preeclampsia: 28.5 ± 5.6. Normotensive with diabetes: 29.9 ± 3.8. Normotensive and non-diabetic: 31.4 ± 4.5.	Vitamin D deficiency was more common in women with T1DM but no concentration predicted preeclampsia. At visit 2 and 3, the 1,25(OH)2D concentration was related with preeclampsia in women with T1DM such as lower concentrations of VDBP	Yes
Mirzakhani, 2018	Boston Massachusetts, Missouri, California	VDAART study	18–39	816	September 2009–July 2011	Group 1: 4000 IU of vitamin D/day + multivitamin with 400 IU vitamin D; group 2: placebo + multivitamin with 400 IU vitamin D	Group 1: 4000 IU of vitamin D/day + multivitamin with 400 IU vitamin D; group 2: placebo + multivitamin with 400 IU vitamin D	Asthma group: n = 327. No asthma group: n = 489	Asthma group: 25.67 ± 5.45 No asthma group: 27.79 ± 5.49	+PE+asthma: n = 29 +PE-asthma: n = 38 Statistical difference among women with controlled and uncontrolled asthma, Vitamin D supplementation reduced risk of preeclampsia by controlling the asthmatic condition	Yes

SBP = Systolic Blood Pressure, DBP = Diastolic Blood Pressure, T1 = Trimester 1, T2 = Trimester 2, T3 = Trimester 3, PE = Preeclampsia, 25(OH)D = 25-hydroxyvitamin-D, CB = Cord Blood, SGA = Small Gestational Age, Treg = Regulatory T cells, PET = Preeclampsia, PMC = Preeclampsia: Mechanisms and Consequences.

## Data Availability

Not applicable.

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
