# Peer review of "Vitamin D Deficiency as a Risk Factor of Preeclampsia during Pregnancy"

_diseases, 2023, doi:10.3390/diseases11040158_

Round 1

Reviewer 1 Report

Comments and Suggestions for Authors

Dear Editor,

I carefully read the manuscript "VITAMIN D DEFICIENCY AS A RISK FACTOR OF PREECLAMPSIA DURING PREGNANCY".

My comments and suggestions for the authors are the following:

 - Linea 274-275: Why was the search conduct by including articles published between 2018 and today? I warmly suggest to the authors to include in their review articles published between inception to today. Moreover, they should specify which date corresponds to "today", for clarity.

 - The authors performed a systematic review of the literature. This point should be clearly specified in the manuscript, in the abstract and in the title too. Moreover, the authors should refer to PRISMA guidelines (and they should also include the guidelines among the references, of course).

 - Line 296: Replace "enrolled studies" with "included studies".

 - The authors should highly consider to refer to doi: 10.3390/nu12020378 in discussion section of the manuscript.

 - English language needs to be carefully revised and improved.

 - Table 1: In the first column, please include the name of the first author of each study.

 - Tables: All the abbreviations used in the tables should be defined at the bottom of the tables.

Comments on the Quality of English Language

Please, see my comments above.

Author Response

Pleas see the attachment.

Reviewer 2 Report

Comments and Suggestions for Authors

This manuscript provides a comprehensive review regarding the role of vitamin D deficiency in pre-eclampsia.

The only concern I have is the use of Google Scholar to find sources to include in the review. This could have led to the inclusion of non-peer-reviewed manuscripts. Have the authors controlled this risk? If this was not possible, manuscripts without a clear and adequate scientific standard should be excluded from the study.

Round 2

Reviewer 1 Report

Comments and Suggestions for Authors

Dear Editor,

I carefully read the manuscript, that is significantly improved compared to the original version.

Reviewer 2 Report

Comments and Suggestions for Authors

My previous comments were met and the manuscript is now clearly improved.